# The Genome Regions Associated with Abiotic and Biotic Stress Tolerance, as Well as Other Important Breeding Traits in Triticale

**DOI:** 10.3390/plants12030619

**Published:** 2023-01-31

**Authors:** Gabriela Golebiowska-Paluch, Mateusz Dyda

**Affiliations:** Institute of Biology, Pedagogical University of Cracow, 30-084 Kraków, Poland

**Keywords:** triticale, doubled haploids, mapping population, quantitative trait loci (QTL), candidate genes, drought, freezing, fungal infection, seedling morphology, yielding capacity

## Abstract

This review article presents the greatest challenges in modern triticale breeding. Genetic maps that were developed and described thus far, together with the quantitative trait loci and candidate genes linked to important traits are also described. The most important part of this review is dedicated to a winter triticale mapping population based on doubled haploid lines obtained from a cross of the cultivars ‘Hewo’ and ‘Magnat’. Many research studies on this population have focused on the analysis of quantitative trait loci regions associated with abiotic (drought and freezing) and biotic (pink snow mold and powdery mildew) stress tolerance as well as related to other important breeding traits such as stem length, plant height, spike length, number of the productive spikelets per spike, number of grains per spike, and thousand kernel weight. In addition, candidate genes located among these regions are described in detail. A comparison analysis of all of these results revealed the location of common quantitative trait loci regions on the rye chromosomes 4R, 5R, and 6R, with a particular emphasis on chromosome 5R. Described here are the candidate genes identified in the above genome regions that may potentially play an important role in the analysis of trait expression. Nevertheless, these results should guide further research using molecular methods of gene identification and it is worth extending the research to other mapping populations. The article is also a review of research led by other authors on the triticale tolerance to the most current stress factors appearing in the breeding.

## 1. Introduction

Triticale (× *Triticosecale* Wittmack), a man-made cereal species, developed by crossing wheat (*Triticum aestivum* L.) and rye (*Secale cereale* L.) was first described in 19th century by Scottish botanist A. Stephen Wilson [1]. The hybrid created by this crossing was originally an octoploid with AABBDDRR genome with a base genomic construction x = 7 [2]. Initially, this crop was created as a species that combined new agronomic, morphological, and utility features [1]. Over time, various types of triticale with different ploidy levels and chromosomal constitutions have been created and evaluated, and currently, a cultivated hexaploid triticale that belongs to the *Poaceae* family contains a genomic constitution of 2*n* = 6x = 42 with the AABBRR genome [3]. It combines valuable traits such as high fertility and grain quality received from wheat with higher stress tolerance obtained from rye [1,4,5]. Up to now, triticale exhibits better drought [6], aluminum [7], freezing [8] as well as waterlogging [9] tolerance than wheat. All of these qualities make triticale a valuable and well-established crop that is cultivated in many European countries such as Poland, Germany, Spain, and France [10]. Triticale grain is mainly used as an animal food, but is also used for human food consumption as well as in bioethanol and biofuel production [3,11,12]. 

This crop is also a valuable genetic bridge for transferring eligible genes among the rye and wheat genetic pool using molecular breeding techniques [4,13,14]. Quantitative trait loci (QTL) mapping, marker-assisted selection (MAS) technique, genomic selection (GS), and next-generation sequencing (NGS) are widely used in improving crop species, especially wheat and rye; in contrast, molecular breeding in triticale is still limited [1,15,16].

Triticale in European cultivations was once fully resistant to many biotrophic diseases such as stem, leaf, and yellow rust as well as *Fusarium* head blight (FHB) and powdery mildew [17]. However, as the sowing areas increased, triticale started to lose its immunity due to the evolution of new pathogen races that can infect this crop [17]. Hence, major breeding objectives are focused on increasing tolerance/resistance to many stress factors simultaneously.

This review paper summarizes the current progress and challenges in modern triticale breeding. The availability of genetic maps together with the QTL regions and candidate genes associated with many important traits that have been identified thus far are described in detail. Additionally, the triticale response to low temperature, fungal infection, drought, and microspore embryogenesis is presented. Moreover, a detailed description of the DH ‘Hewo’ × ‘Magnat’ mapping population’s importance is introduced as a guide for further research.

## 2. Genetic Maps

Presently, molecular biology techniques, mostly methods based on molecular markers, are important tools used in modern plant breeding. A creation of high-density reference genetic maps of entire genomes is an indispensable part that can be widely used in marker-assisted selection (MAS) and genomic selection (GS) [1]. The main advantage of a genetic map is that it can provide a useful resource for comparative genomics, the mapping of quantitative trait loci (QTL) associated with multiple important traits as well as linking physical and genetic maps, and consequently, detecting candidate genes associated with multiple proteins. High-density genetic maps have been described for many plants including crops such as wheat, rye, and barley but also, for triticale [15,16,18,19,20,21,22,23]. Most of those maps were developed using DArT (diversity arrays technology), DArT-seq (diversity arrays technology sequencing) and SNP (single-nucleotide polymorphism) marker systems, which are widely used in genetic map construction for multiple crop species.

The first triticale genetic map was created by using 73 doubled haploid (DH) lines derived from F_1_ plants that originated from a cross between cv. ‘Torote’ and cv. ‘Presto’ [18]. This map was 2465.4 cM in length and contained in total 356 markers assigned to 21 linkage groups. After Badea et al. [14] reported the development of a triticale-specific DArT array combining markers developed in wheat, rye, and triticale, DArT was successfully implemented in a linkage map creation. Alheit et al. [19] described a more advanced, consensus triticale genetic map derived from nine parental lines. This map was 2309.9 cM in length and composed of 2555 DArT markers assigned to 22 linkage groups, seven for the A and B subgenomes as well as eight for the R subgenome (chromosome 2R is composed of two linkage groups 2R-1 and 2R-2).

Tyrka et al. [20] constructed a genetic map for the DH ‘Saka3006’ and ‘Modus’ population. This map contains 1568 markers (1385 of them are DArT markers) assigned to 21 linkage groups with the total map length of 2397 cM. Subsequently, along with the development of DArT-seq technology, Tyrka et al. [21] used this kind of molecular marker to create a linkage map for a population of 89 DH ‘Hewo’ × ‘Magnat’ lines. This map was 4907.4 cM in length and composed of 3593 markers assigned to 20 linkage groups. Additionally, another consensus map of six DH populations was described by Tyrka et. al. [15] with a 4593.9 cM length consisting of 1576 unique DArT markers (3086 markers in total).

The genetic map for the F_2_ ‘Lamberto’ × ‘Moderato’ population was reported by Karbarz et al. [22]. This map was composed of 911 markers and assigned to 21 linkage groups with a total map length of 2837.3 cM. Subsequently, Wąsek et al. [23] described a modified genetic map for the ‘Hewo’ × ‘Magnat’ DH population. This map was 1367.7 cM long and composed of 41 SSR and 680 DArT markers ordered into 22 linkage groups (the 7A, 2B, and 3B chromosomes were represented by double linkage groups). This map had a higher mean map density (4.7) compared to the previous one (2.8; [21]). The most recent genetic map for triticale was described by Dyda et al. [16]. A genetic map for 168 lines of the ‘Grenado’ × ‘Zorro’ DH population was created mainly based on DArT-seq and DArT markers. This map was composed of 1891 unique markers ordered to 21 linkage groups with a total map length of 5249.9 cM and marker saturation of 2.8. 

When comparing the total marker number of all triticale genetic maps, the results were very different. The highest number of markers assigned to the A and B linkage group was found in the DH ‘Hewo’ × ‘Magnat’ line mapping population [21] while for the R group in the DH ‘Saka’ × ‘Modus’ line mapping population [20]. Additionally, the A group was previously described by Tyrka et al. [20,21], Karbarz et al. [22], Wąsek et al. [23], and Dyda et al. [16] as the one with the lowest number of markers assigned, regardless of the marker type used in the map construction.

## 3. QTL and Candidate Gene Analysis

As mentioned, genetic maps can be used in positioning quantitative trait loci (QTL) that are linked with multiple traits. Up to now, many research studies have described the QTL regions associated with triticale resistance to both biotic and abiotic stresses such as resistance to *Fusarium* head blight (FHB) [17,24,25,26], powdery mildew [16,22,27,28], yellow rust [11,17,29], pink snow mold [30,31,32], drought [33,34,35], and freezing [23,32]. Many studies showed the analysis of QTL regions associated with the important agronomic factors and morphological features of triticale [36,37,38,39,40]. Additionally, from the breeding perspective, QTL regions linked with androgenic responsiveness [41], albino plant formation [42], and ABA accumulation in the anthers in response to stress factors [43,44] were also identified, which can be very useful in modifying an in vitro approach, especially in the androgenesis process.

Knowledge of the position of the genomic regions linked with an important trait may help with candidate gene identification. Such genes that encode multiple proteins can be widely used in modern molecular breeding programs, but so far, not many studies have described such genes identified in crops. In triticale, the information of important candidate genes is still limited [16,22,23,28,32,45].

## 4. Low Temperature Tolerance

The ability of cereal seedlings to survive the winter (i.e., winter hardiness) depends on the plant’s ability to tolerate a wide range of environmental stresses such as freezing, changing temperatures and climate, low light intensity, desiccation, wind, snow cover, icing, and various winter-related diseases [46].

There is a large variation in triticale freezing tolerance. A lower/higher level of this feature may be related to the genetic hexaploid structure, where part of the chromosomes of the D wheat subgenome and R rye subgenome contains genes important for winter hardiness. Thus, selection for freezing tolerance in triticale seems to be very important for further expansion of this crop area. A strong interaction between the frost tolerance of the genotype and the environment was also observed in triticale [23]. Freezing induces complex and gradual changes in the photosynthetic apparatus [47,48]. In winter, the photosynthetic apparatus is damaged not only by freezing, but also by the photoinhibition of photosynthesis. Tolerance to cold-induced photoinhibition appears to be closely related to freezing tolerance, partly due to the common mechanisms of acclimatization to both stresses [49].

Temporary warming in winter can cause a reduction in snow cover and deacclimatization, ultimately reducing winter hardiness [46,50]. The risk of frost infestation is related to the number of days with a daily minimum temperature below the set freeze tolerance level (−20 °C) on days without continuous snow cover [8].

As has been shown by many years of research, low temperature may increase the triticale tolerance to stresses that coexist with cold in winter, which could remain persistent during the following early spring period and even at the plant adult stage. Tolerance to freezing [23] and pink snow mold caused by *Microdochium nivale* [32] appears after a cold-hardening period and it is an essential, genotype-dependent, complex quantitative trait for wintering.

## 5. Fungal Infection Tolerance after Cold-Acclimation

Tolerance to low temperature and diseases caused by the fungi infecting seedlings in the cold is an essential trait for triticale overwintering. Long-term studies indicate that after long-term exposure to low temperature (acclimation, hardening), cereal seedlings acquire genotype-dependent cross-tolerance to other later stresses. Low temperature (4 weeks at 4 °C) may increase the tolerance of triticale seedlings to stresses coexisting with cold in winter, which may persistently follow throughout the early spring period and even in the adult phase of the plant [16,23,28,32,51,52]. Tolerance to pink snow mold caused by *M. nivale* appears after a period of hardening with low temperature and is a genotype-dependent, complex feature significant for wintering seedlings [23,31,32,52,53]. Furthermore, Dyda et al. [31] presented new insights into the mechanism of triticale resistance to *M. nivale*. Their experiment with three different *M. nivale* strains and three different infection assays showed that plants that maintained a higher maximum quantum efficiency of PSII showed, at the same time, less leaf damage upon infection. *M. nivale* can establish necrotrophic or biotrophic interactions with the susceptible or resistant genotypes, respectively. The genetic regions associated with PSII functioning and resistance, together with a wide range of PSII- and resistance-related genes, were found on chromosomes 4 and 6. In addition, it was confirmed that the structural and functional integrity of the plant are required factors to meet the energy demand of infected cells, photosynthesis-dependent systemic signaling, and defense responses [31].

The proteomic profiling allowed for the identification of candidate proteins associated with the cold acclimation of triticale seedlings [53] as well as the tolerance to freezing and pink mold infection [52]. The content of individual proteins was analyzed by two-dimensional gel electrophoresis (2-DE) and matrix-enhanced time-of-flight laser desorption/ionization (SELDI-TOF). Low temperature exposure of seedlings only caused quantitative changes in the leaves of both cultivar parents, causing an increase or decrease in the abundance of the proteins with a molecular weight of 4–50 kDa. Among the proteins accumulated under the influence of cold in the leaves of the tolerant cultivar ‘Hewo’, two thioperoxidases (antioxidant proteins specific for chloroplastic thiols) as well as a subunit of mitochondrial ATP synthase and ADP-binding resistance protein were identified [53]. On the other hand, in low-temperature hardened seedlings of this genotype, a reduced level of the small subunit RuBisCO and the PW9 subunit of peroxidase 10 was observed. Simultaneous SELDI-TOF analysis revealed several proteins of low weight with increased concentration in cold-exposed plants of the tolerant genotype versus the sensitive one. Non-gel protein profiling in triticale seedlings was performed by high-performance liquid chromatography coupled with mass spectrometry (LC-MS) and Raman spectroscopy [52]. Seedlings of doubled haploid (DH) lines selected from the ‘Hewo’ × ‘Magnat’ mapping population with extreme tolerance/susceptibility to freezing and *M. nivale* infection were used in these studies. These untargeted methods led to the detection of twenty-two candidate proteins that accumulated under the influence of low temperature in the most tolerant seedlings in relation to the susceptible ones, classified as biomolecules involved in protein biosynthesis, response to various stimuli, energy balance, response to oxidative stress, protein modification, membrane construction, and anthocyanin synthesis. Additionally, in seedlings of the most frost-tolerant line and *M. nivale*, hardening resulted in a decrease in the content of carotenoids and chlorophyll. Moreover, a decrease was detected in the intensity of the spectra characteristic for carbohydrates and an increase in the intensity spectra characteristic of the protein compound. Both tested lines—tolerant and sensitive to freezing and *M. nivale* infection—showed different stress responses in the characteristic phenolic components [52].

Żur et al. [54] showed that the ‘Hewo’ and ‘Magnat’ response to cold treatment and *M. nivale* infection affected the accumulation of b-1,3-glucanase and chitinase. The results indicate that both glucanhydrolases were substantially suppressed as the result of cold treatment in both cultivars due to altered metabolism. On challenge with *M. nivale*, ‘Hewo’ showed a marked increase in chitinase while none of the cultivars showed a change in glucanase, confirming the role of chitinases in resistance against *M. nivale* [54]. Gawrońska and Gołębiowska [55] identified biochemical markers that are potentially involved in increased resistance against *M. nivale*. It was shown that the triticale genotype and seedling treatment may influence the level of TBARS, which is well-known as a marker of oxidative stress in response to different abiotic and biotic factors. Plant resistance after cold hardening [56], cold-enhanced gene expression at seedling stage [57], cold-modulated small protein abundance at the seedling stage [53], presence and concentration of free and cell wall-bound phenolic acids [51], cold-modulated leaf compounds [52], cold-induced changes in cell wall stability [58], and changes in the physical and chemical leaf properties [59] were tested on the ‘Hewo’ × ‘Magnat’ population after *M. nivale* infection and provided new insights on triticale cold-acclimation as well as plant vs. pathogen interactions.

In addition to the aforementioned pink snow mold tolerance studies, Karbarz et al. [22] and Dyda et al. [16,28] described triticale adult-plant resistance to *B. graminis* infection measured at the adult plant stage in the field after isolate mixture inoculation and after natural field infection, accordingly. Using the area under disease progress curve (AUDPC), maximum disease severity (MDS), and the average value of powdery mildew infection (avPM) methods, many new QTL regions and candidate genes associated with powdery mildew resistance were described for the first time and presented. Such regions, after careful validation in available triticale varieties and lines, can be used in marker-assisted selection (MAS) and the pyramiding of adult-plant resistance (APR) genes to PM in triticale breeding that can assist molecular breeding programs.

## 6. Importance of the DH ‘Hewo’ × ‘Mangat’ Mapping Population

Our meta-analysis was mainly focused on the ‘Hewo’ × ‘Magnat’ mapping population composed of 89 doubled haploid (DH) lines that were derived by the anther method described in detail by Wędzony [60]. All lines were derived from the F_1_ hybrid of two triticale cultivars, ‘Hewo’ (Strzelce Plant Breeding, IHAR Group Ltd., Poland) was used as a female parent and ‘Magnat’ (DANKO Plant Breeders Ltd., Poland). Both parental cultivars showed a different response to multiple biotic and abiotic factors.

Over the years, the DH ‘Hewo’ × ‘Magnat’ line population has been used as a model to test various traits and plant responses at different stages of development and under both natural and controlled conditions. These studies were conducted at the anatomical, physiological, biochemical, and/or genetic levels. The first genetic map constructed for the DH ‘Hewo’ × ‘Magnat’ lines enabled a new approach of research on this population. The map was 4997.4 cM long and composed of 3539 markers in total (842 DArT, 2647 SNP-DArT, and 50 SSR markers), which were ordered into 20 linkage groups assigned to the A (7), B (7), and R (6) subgenomes [21].

The results showed the quantitative trait loci (QTL) associated with many traits measured in natural and controlled conditions at the seedling and adult plant stage. Tolerance to drought [34,35], *M. nivale* infection [31,32], and freezing [23] were tested at the seedling stage after vernalization in a cool chamber at 4 °C under controlled conditions. Additionally, tolerance to the powdery mildew caused by *Blumeria graminis* [28] was tested in natural field conditions at the adult stage and morphological traits such as stem length, plant height, spike length, number of productive spikelets per spike, number of grains per spike, and thousand kernel weight [38] were determined at the adult plant stage along with the identification of QTL. Additionally, for the above mappings, multiple candidate genes were identified and described for the first time.

## 7. The Comparative Analysis of the Genomic Results Obtained from the DH ‘Hewo’ × ‘Magnat’ Mapping Population

In subgenome A, the QTL regions on four wheat-derived chromosomes were identified; the trait loci were located in different regions of chromosomes 2A, 4A, 5A, and 7A (Figure 1). In subgenome B, the QTL regions on six wheat-derived (1B, 2B, 3B, 4B, 6B, and 7B) chromosomes were identified and the trait loci were located in different regions (Figure 2). In subgenome R, the QTL regions on six rye-derived chromosomes (1–6R) have been identified. The trait loci were located in different regions of 1R, 2R, and 3R chromosomes (Figure 3). QTL regions common to two or more traits have been detected on 4R, 5R, and 6R (Figure 3). The common region for the greatest number of traits was identified on 5R between 0 and 37 cM (Figure 4). According to the flanking marker sequences, this region was estimated between 865 109 880 and 874 163 401 kb (9 Mb long). The QTL regions, together with the candidate genes related to the studied traits of the DH ‘Hewo’ × ‘Magnat’ mapping population, are presented in Table 1.

In the 5R region common for different QTL, identified between 0 and 37 cM of this chromosome, candidate genes were in silico identified here with the method described previously [32]. All candidate genes had a positive additive effect from cv. ‘Hewo’ and included six candidate records encoding the predicted (1) chloroplastic FAF-like protein; (2) F-box/FBD/LRR-repeat protein; (3) Myosin-10-like protein; (4) thylakoid membrane protein TERC; (5) xyloglucan endotransglucosylase/hydrolase; and (6) nucleotide-gated ion channel protein (Table 2). The first of the proteins listed above is associated with a negative regulation of ABA-activated signaling pathway as well as the positive regulation of phosphatase activity. The second is involved in the regulation of short-day photoperiodism and flowering. Another gene potentially related to the studied traits of the mapping population encodes the thylakoid membrane protein TERC, an integral protein that plays a crucial role in thylakoid membrane biogenesis and thylakoid formation in early chloroplast development, essential for the synthesis of photosystem II (PSII) core proteins and is required for the efficient insertion of thylakoid membrane proteins.

Other candidate genes located in a section of chromosome 5R in the rye subgenome database were identified here and include over sixty genes involved mainly in the immune response to abiotic and biotic stimuli, signaling, and oxidoreductase activity (Table 3). At least twenty-four of them are described in the literature as coding proteins associated with the response/tolerance to abiotic and biotic stresses: 2-oxoglutarate-dependent dioxygenase, ankyrin repeat-containing protein, ATP-dependent RNA helicase DeaD, beta-amylase, BTB/POZ and MATH domain-containing protein, chaperone DnaK, cold regulated protein (COR), cysteine proteinase inhibitor, embryogenesis transmembrane protein-like, F-box family protein, flavin-containing monooxygenase, heat shock transcription factor, homeobox leucine-zipper protein, lipid transfer protein, L-type lectin-domain containing receptor kinase VIII, metacaspase-1, NAC domain-containing protein, nascent polypeptide-associated complex subunit beta, peptidoglycan-binding LysM domain protein, peroxidase, photosystem I assembly protein Ycf3, polyubiquitin, rRNA N-glycosidase, and terpene synthase. COR proteins are also known as involved in the vegetative to reproductive phase transition of meristem. Many proteins from those listed in Table 3 such as cyclin delta-3, cysteine proteinase inhibitor, embryogenesis transmembrane protein-like, F-box family protein, folylpolyglutamate synthase, polyamine oxidase, protein FAR1-RELATED SEQUENCE 5, ribosomal RNA small subunit methyltransferase J, senescence regulator (DUF584), WD repeat-containing protein, and xyloglucan 6-xylosyltransferase are reported as involved in seed development. Next, cytochrome P450 identified here is described as associated with the regulation of growth as well as leaf and root development. The other candidate proteins identified here may contribute to transcription regulation: basic helix-loop-helix (bHLH) DNA-binding superfamily protein, Basic-leucine zipper (bZIP) transcription factor family protein, cold regulated protein (COR), histone acetyltransferase of the CBP family 12, homeobox leucine-zipper protein, LURP-one-like protein, NAC domain-containing protein, protein FAR1-RELATED SEQUENCE 5, small nuclear ribonucleoprotein, THO complex subunit 1, transcription elongation factor GreA, and Trihelix transcription factor GT-2. Many other candidate genes may encode proteins involved in response to hormones such as 2-oxoglutarate-dependent dioxygenase, basic helix-loop-helix (bHLH) DNA-binding superfamily protein, ankyrin repeat-containing protein, cold regulated protein (COR), cyclin delta-3, embryogenesis transmembrane protein-like, F-box family protein, homeobox leucine-zipper protein, lipid transfer protein, P1, polyamine oxidase, and polyubiquitin (Table 3).

Regions with a similar position in terms of powdery mildew tolerance in the two mapping populations were identified on chromosomes 4A, 5R, and 6R. Powdery mildew tolerance region Qpm.gz.5R.1 in the 168 DH 'Grenado' × 'Zorro' line population was flanked by markers 4357257 and 4218107 (95.2–109.7 cM), 4357414 and rPt-401500 (45.7–60.5 cM) as well as 4352431 and 4348906 (0.0–34.6 cM) on the 5R chromosome, as described in Dyda et al. [28]. Similarly, in the DH ‘Hewo’ × ‘Magnat’ population, 5R region for the powdery mildew field tolerance was identified between 34.6 and 37.0 cM. Moreover, similar 6R regions (55.8–60.2 cM and 203.2–209.4 cM) as well as a 4A region (103.2–111.4 cM) on the DH ‘Hewo’ × ‘Magnat’ QTL map were identified for powdery mildew field tolerance in comparison to the DH ‘Grenado’ × ‘Zorro’ QTL map.

In summary, the presented results indicate the location of common QTL regions on the 4R, 5R, and 6R rye chromosomes, with particular emphasis on 5R. In silico localized candidate genes potentially play important roles in the trait's expression. Nevertheless, these results should guide further research using molecular gene identification methods and it is worth extending research to other mapping populations. 

## 8. Drought Tolerance

Soil drought can significantly accelerate the aging of plants by a gradual decrease in metabolic activity, ultimately leading to cell death [35]. Gelang et al. [61] associated the decrease in plant yield with accelerated senescence caused by soil drought and a significant shortening of the grain filling phase. The first visible sign of aging is the yellowing of the leaves as a result of chlorophyll degradation and the appearance of other dominant pigments, mainly carotenoids, xanthophylls, and anthocyanins. Chlorophyll a decays faster than chlorophyll b during aging, leading to a decrease in the chlorophyll a/b ratio [62]. It has also been observed that the degradation of the reaction centers precedes the degradation of the proteins that make up the light-harvesting complex in PSII. Another factor that distinguishes the stay-green genotypes is the increase in the carbohydrate content of the green parts compared to the normal aging genotypes. 

Increased accumulation of soluble carbohydrates in long-lasting green plants is often accompanied by increased leaf assimilation area in the grain-filling phase [61]. A higher yield of plants with delayed aging may result, for example, from maintaining a high level of soluble carbohydrates in the leaves. Carbohydrates are used in the synthesis of phenolic compounds involved in plant defense during environmental stresses, and can also serve as indicators of plant aging. In aging plant organs, the level of phenolic compounds increases at the expense of soluble carbohydrates. This indicates the important role of sugars in the integration of environmental signals during the regulation of leaf senescence [63]. 

Mechanisms related to the triticale aging in conditions of soil drought can be controlled by the genome of wheat and/or rye; however, triticale also does not show the specific drought responses that neither wheat nor rye have [34,64]. The genetic and molecular basis of triticale acclimatization to drought has so far been poorly understood. It is not clear whether triticale responses to drought are specific to the wheat or rye genome or result from the activity of both subgenomes. Cereal aging studies tend to focus on the flag leaf, however, the aging of triticale progresses from its lower parts below the flag leaf, which are the first to show clear signs of drought-induced aging and then run toward the flag leaf [64].

## 9. Effective Microspore Embryogenesis

Another important breeding feature of triticale associated with the effect of low temperature is the ability to undergo a process of somatic microspore embryogenesis (ME)–androgenesis, used for the development of new plant lines from single, unfertilized pollen grains or from the anthers. Effective ME requires significant modifications in the pattern of gene expression, followed by changes in the cell's proteome and metabolism. Recent research has also aroused an interest in the role of epigenetic factors in de-differentiation and reprogramming the microspores to develop into an embryo. Therefore, a demethylating agent (2.5–10 μM 5-azacitidine, AC) along with low temperature (3 weeks at 4 °C) for ME induction was used for two doubled haploid triticale lines selected from the ‘Saka 3006’ × ‘Modus’ population, and their effect was analyzed in relation to the protein profiles of pollen grains as well as the expression of selected genes (TAPETUM DETERMINANT1 (similar to TaTPD1), SOMATIC RECEPTOR KINASES EMBRYOGENESIS 2 (SERK2), and GLUTATHIONE S-TRANSFERASE (GSTF2) as well as the efficiency of ME [65]. The use of a concentration of 5.0 μM AC was the most effective for ME induction; this was associated with the inhibition of intensive anabolic processes, mainly photosynthesis and light-dependent reactions, transition for effective catabolism and mobilization of carbohydrate reserves to satisfy the high energy demand of cells during microspore reprogramming and effective defense against stress-inducing effects (i.e. protection of the correct one folding during protein biosynthesis and efficient degradation of dysfunctional or damaged proteins). Additionally, the use of a demethylating agent at a concentration of 5.0 μM AC enhanced the expression of all genes previously identified as related to the embryogenic potential of microspores (i.e. similar to TaTPD1, SERK, and GSTF2) [65]. 

The effectiveness of ME is determined by a complex network of internal and environmental factors. In the presented tests of triticale, a strong positive correlation between the generation of hydrogen peroxide and ME efficiency confirmed the important role of reactive oxygen species in microspore reprogramming toward somatic embryogenesis [66]. However, for high efficiency ME induction, intensive hydrogen peroxide production must be associated with high activity antioxidant enzymes, superoxide dismutase and catalase. As revealed in a study, a strong seasonal influence on the physiological state of microspores suggests a kind of “biological clock” controlling plant reproduction, crucial for the viability of microspores and embryogenic potential. Although the impact of various modifications of plant material pre-treatment causing stress inducing ME was determined mainly by condition microspores, but with higher stress intensity (3 weeks at 4 °C), positive effects induced by antioxidant molecules were observed for reduced glutathione and its precursor, 1-2-oxothiazolidine-4-carboxylic acid. A high level of variability in response to the inducing initial ME was also demonstrated in the processing of the material between two DH lines of triticale and among microspores isolated from later-developing spikes [66].

## 10. Conclusions

The biggest challenges in modern triticale breeding are abiotic stresses such as drought and freezing as well as biotic stresses caused by fungal pathogens. In response to this demand, several genetic maps have been developed and described so far, together with the quantitative trait loci and candidate genes linked to important triticale traits. For many years, studies conducted on a winter triticale mapping population based on doubled haploid lines obtained from a cross of cultivars ‘Hewo’ and ‘Magnat’ focused on the analysis of quantitative trait loci regions associated with abiotic and biotic stress tolerance as well as related to other important breeding traits. A comparison analysis of those results revealed the location of common quantitative trait loci regions on the rye chromosomes 4R, 5R, and 6R, with particular emphasis on chromosome 5R. The most valuable are the QTL regions that have been repeated in different experiments as well as in different locations and years. Less reliable are the QTL regions obtained in a single experiment. As described in this paper, the candidate genes identified in the above genome regions may potentially play an important role in analyzing trait expression. Nevertheless, these results should guide further research using molecular methods of gene identification and it is worth extending the research to other mapping populations. 

## Figures and Tables

**Figure 1 plants-12-00619-f001:**
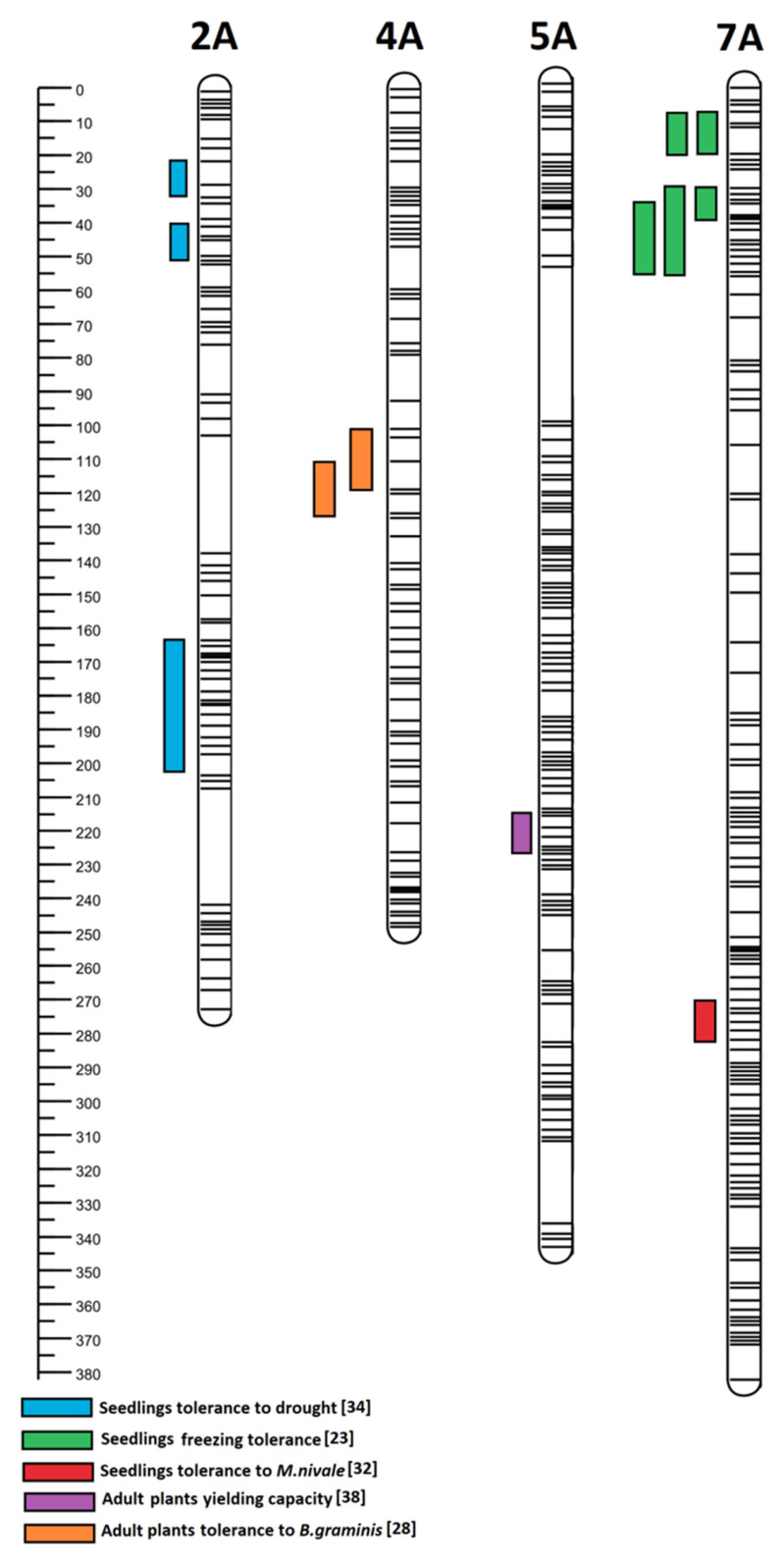
Subgenome A regions identified for the winter triticale DH ‘Hewo’ × ‘Magnat’ mapping population, related to the seedling tolerance to drought, freezing, and pink snow mold caused by *M. nivale* as well as the tolerance of adult plants to powdery mildew and yielding capacity in the generative phase. Subgenome regions were determined by bioinformatics using the genetic map published by Tyrka et al. [21] and Windows QTL Cartographer V2.5_011 [23,28,32,34,38].

**Figure 2 plants-12-00619-f002:**
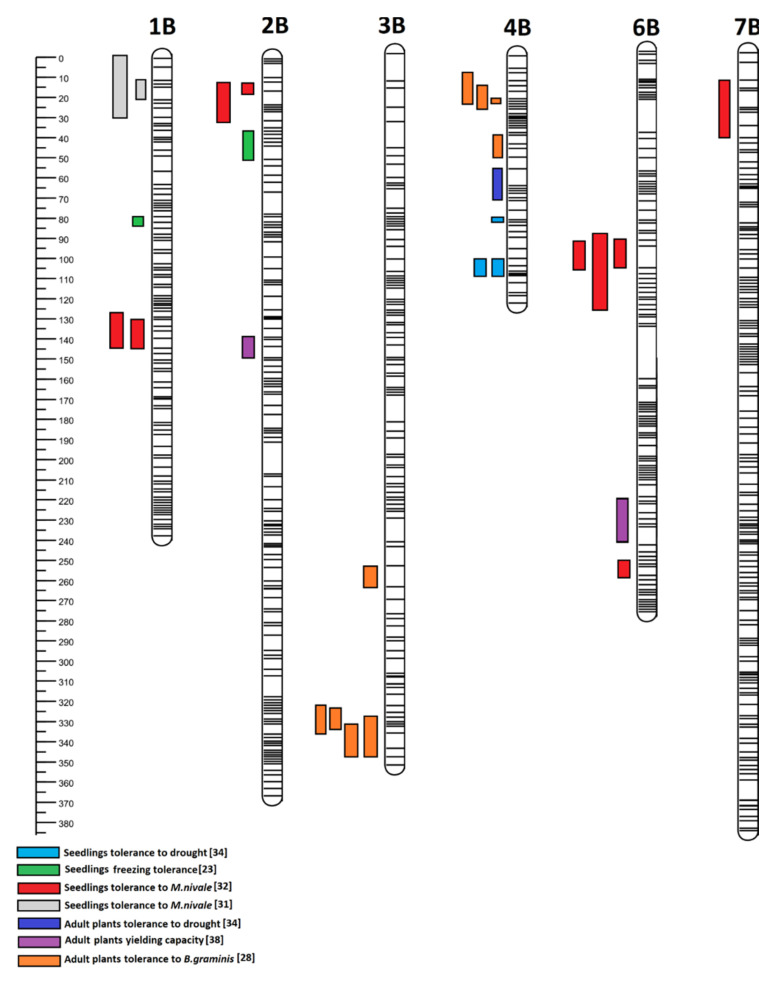
Subgenome B regions identified for the winter triticale DH ‘Hewo’ × ‘Magnat’ mapping population, related to seedling tolerance to drought, freezing and pink snow mold caused by *M. nivale*, as well as adult plants tolerance to drought, powdery mildew and yielding capacity in the generative phase. Subgenome regions were determined by bioinformatics using the genetic map published by Tyrka et al. [21] and Windows QTL Cartographer V2.5_011 [23,28,31,32,34,38].

**Figure 3 plants-12-00619-f003:**
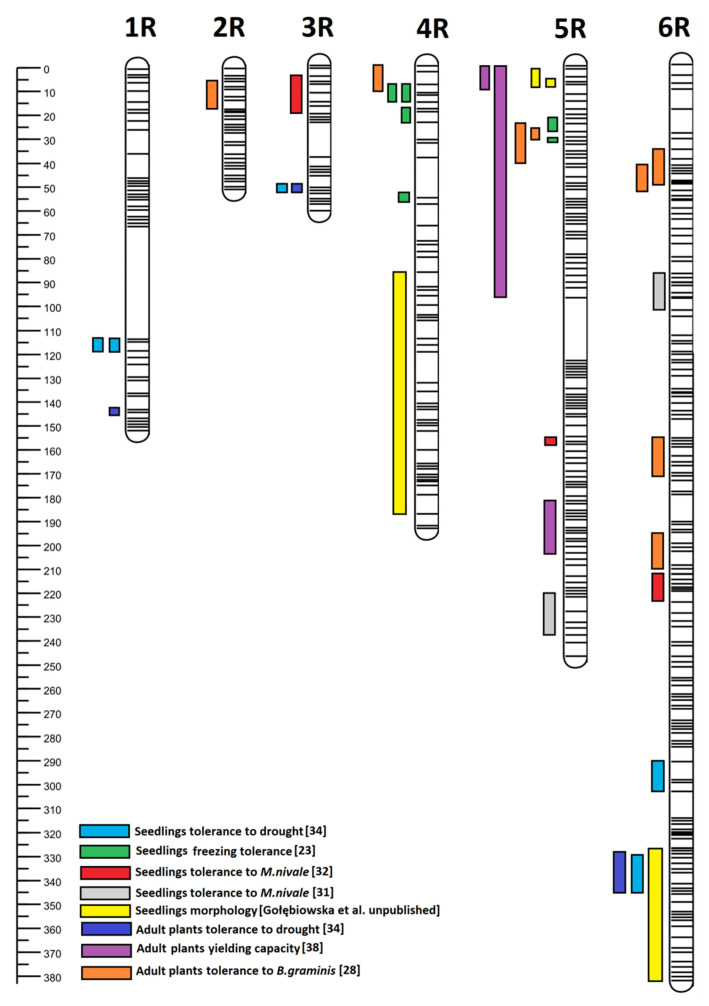
Subgenome R identified for the winter triticale DH ‘Hewo’ × ‘Magnat’ mapping population, related to seedlings the tolerance to drought, freezing, and pink snow mold caused by *M. nivale* as well as the tolerance of adult plants to drought, powdery mildew, and yielding capacity in the generative phase. Subgenome regions were determined by bioinformatics using the genetic map published by Tyrka et al. [21] and Windows QTL Cartographer V2.5_011 [23,28,31,32,34,38].

**Figure 4 plants-12-00619-f004:**
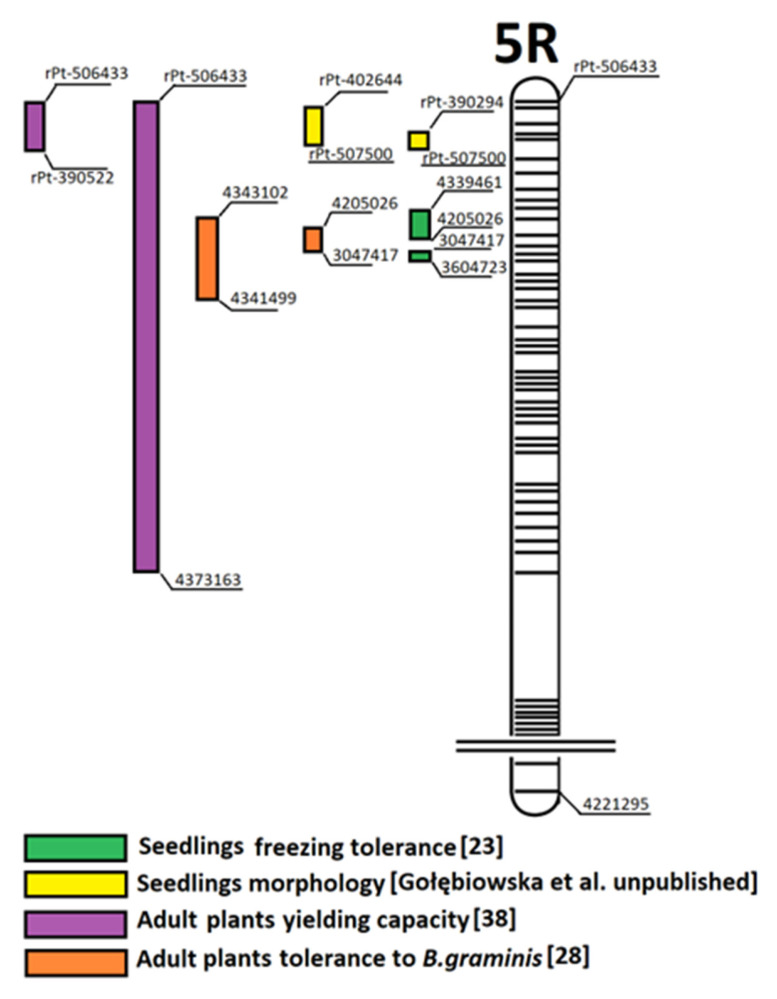
The 5R rye chromosome region containing regions determined here on the basis of experimental and bioinformatic studies, related to the freezing tolerance of winter triticale seedlings and their morphology parameters as well as the tolerance of adult plants to powdery mildew in the field and the yielding capacity in the generative phase [23,28,38].

**Table 1 plants-12-00619-t001:** Identified QTL regions, along with candidate genes in these regions and their predicted function, associated with abiotic and biotic stress tolerance as well as other important breeding traits in triticale.

Trait	Chromosome	QTL Position [cM]	ID of the Candidate Sequence Identified	Predicted Protein Coded by the Candidate Sequence	Predicted Function of the Candidate Protein	Additive Effect *	Reference
**Seedling morphology after cold-acclimation (controlled conditions)**							
Second leaf length, second leaf sheath length, second leaf blade length as well as second leaf length to second leaf width ratio	5R	5–11	a. XM_044505887.1 b. XM_044528234.1 c. XM_020293712.3	a. F-box/FBD/LRR-repeat protein b. Myosin-10-like c. Thylakoid membrane protein TERC	a. Negative regulation of long-day photoperiodism and flowering; regulation of short-day photoperiodism and flowering b. ATP binding myosin complex; cytoskeletal motor actin filament binding c. Integral thylakoid membrane protein that plays a crucial role in thylakoid membrane biogenesis and thylakoid formation in early chloroplast development, essential for de novo synthesis of photosystem II (PSII) core proteins and required for efficient insertion of thylakoid membrane proteins. May assist synthesis of thylakoid membrane proteins at the membrane insertion step	H	Gołębiowska et al., unpublished
Second leaf width	4R	91–193	a. XM_044530858.1 b. XM_044507273.1	a. Pentatricopeptide repeat-containing protein b. Receptor-like cytoplasmic kinase 176	a. Chloroplast RNA modification b. Protein serine/threonine/tyrosine kinase activity; defense response; transferase activity; innate immune response. Participates in the activation of defense genes during response to PGN and chitin		
	6R	325–383	XM_044584412.1	Signal peptide peptidase-like 5	Aspartic endopeptidase activity, intramembrane cleaving; lysosomal membrane; integral component of cytoplasmic side of endoplasmic reticulum membrane; Golgi associated vesicle membrane; membrane protein proteolysis		
Second leaf length to second leaf width ratio	5R	3–11	XM_044528388.1	Xyloglucan endotransglucosylase/ hydrolase	Hydrolase activity, hydrolyzing O-glycosyl compounds; apoplast; cellular glucan metabolic process; xyloglucan: xyloglucosyl transferase activity		
Second leaf length to second leaf width ratio	4R		XM_044588536.1	Fertilization independent endosperm 1 protein	Part of ESC/E(Z) complex in nucleus; nucleosome binding; negative regulation of transcription by RNA polymerase II; chromatin silencing complex; protein binding; chromatin organization; response to cold; negative regulation of flower development; vernalization response; regulation of endosperm development; histone methylation; regulation of gene expression by genomic imprinting; DNA-binding transcription factor activity		
**Seedling freezing tolerance after cold-acclimation (controlled conditions)**							
Electrolyte leakage and recovery after freezing during two years	7A	27.5–38.3				M	[23]
		27.5–57.2	XM_037576333.1	PPR protein	Involved in chloroplast RNA processing, modification and splicing	H	
Electrolyte leakage during two years		8.9–19.9	XM_037604689.1	BTR1-like protein	mRNA binding and regulation of gene expression		
Recovery after freezing	1B	79.9					
Recovery after freezing	2B	36.3–46.1					
Electrolyte leakage during two years	4R	9.8–16.5					
Recovery after freezing during two years		22.6–25.2				M	
		57.2					
	5R	22.2–26.5					
Electrolyte leakage		32.0–32.9	XM_037587241.1	Nucleotide-gated ion channel	Voltage-gated potassium channel activity, cyclic nucleotidegated ion channel, integral component of membrane, transmembrane helical protein	H	
**Seedling tolerance to drought after cold-acclimation (controlled conditions)**							
Soluble phenolics content in in seedling’s leaves under drought stress	2A	26.8–32.5 164.1–202.5	XM_044473408.1	DNA topoisomerase	Chromatin organization and progression of endoreduplication cycles, reciprocal meiotic recombination		[34]
	1R	109.0–115.3					
Cell wall-bound phenolics content in seedling’s leaves under drought stress	4B	107.0–116.1	XM_037571660.1	PLASMODESMATA CALLOSE-BINDING PROTEIN	Able to bind (1->3)-beta-D-glucans (laminarin). Probably involved in cell-to-cell trafficking regulation.		
Seedling’s leaf osmotic potential under drought stress	2A	43.7–52.3					
	1R	101.0–116.3					
	6R	332.8–345.9					
The maximum quantum efficiency of PSII (Fv/Fm) in the seedling’s leaves under drought stress	4B	82.4–82.6					
		106.8–116.1	XM_037571660.1	PLASMODESMATA CALLOSE-BINDING PROTEIN	Able to bind (1->3)-beta-D-glucans (laminarin). Probably involved in cell-to-cell trafficking regulation.		
	3R	56.8–60.1	XM_044496729.1	General transcription factor IIE subunit	Transcription initiation from RNA polymerase II promoter		
	6R	293.7–300.4	XM_044586654.1	Tryptophan decarboxylase	May play a major role in serotonin biosynthesis during senescence. Accumulation of serotonin attenuates leaf senescence.		
**Seedling tolerance to *M. nivale* infection after cold-acclimation (controlled conditions)**							
Infection index as well as tolerance level	1B	128.6–144.1				M	[32]
						H	
Infection index as well as tolerance level	2B	9.3–37.5	XM_044465446.1	Sterol 3-betaglucosyltransferase	Lipid glycosylation; UDP-glycosyltransferase activity; carbohydrate metabolic process	M	
						H	
Infection index as well as tolerance level during two years	6B	93.9–128.3 248.7–254.8				M	
						H	
						M	
Tolerance level	7A	270.1–284.8				H	
Tolerance level	7B	17.6–42.2					
Tolerance level	3R	6.8–22.7					
Tolerance level	5R	155.1–161.3				M	
Tolerance level	6R	213.2–225.2					
The maximum quantum efficiency of PSII (*F_v_/F_m_)* after *Microdochium nivale* infection	1B	10.0–17.0	a. XM_044523565.1 b. XM_037589351.1	a. Leucine-rich repeat receptor-like protein kinase b. Disease resistance protein RGA4	a. Receptor with a serine/threonine-protein kinase activity b. Resistance proteins guard the plant against pathogens that contain an appropriate avirulence protein via a direct or indirect interaction with this avirulence protein. That triggers a defense system which restricts the pathogen growth		[31]
		1.0–34.0					
	6R	88.0–108.0					
**Powdery mildew field tolerance in the adult stage**							
Field tolerance to powdery mildew in two different locations	4A	103.2–125.6	XM_037572659.1	Disease resistance protein RGA3	Resistance proteins guard the plant against pathogens that contain an appropriate avirulence protein via a direct or indirect interaction with this avirulence protein. That triggers a defense system which restricts the pathogen growth	M	[28]
Field tolerance to powdery mildew in two different locations, for two locations the same result in two years	3B	254.4–264.9				H	
		323.9–349.4	Serine/threonine-protein kinase		Contributes to pathogen-associated molecular pattern (PAMP)-triggered immunity (PTI) signaling including calcium signaling and root growth inhibition, and **defense responses** downstream of FLS2		
Field tolerance to powdery mildew in two different locations during two years	4B	15.0–56.2	XM_037572350.1	NAC domain-containing protein	Transcriptional activator that mediates auxin signaling to promote lateral root development. Activates the expression of two downstream auxin-responsive genes		
Field tolerance to powdery mildew in one location	2R	7.8–17.2					
Field tolerance to powdery mildew in one location	4R	0.0–11.3	MG672525.1	*Triticum timopheevii* isolate QPm.tut-4A powdery mildew resistance region genomic sequence	Associated with race non-specificity and incomplete resistance		
Field tolerance to powdery mildew in three different locations	5R	24.2–37.0	XM_037583016.1	Chloroplastic protein FAF-like	Negative regulation of abscisic acid-activated signaling pathway, negative regulation of phosphorylation, positive regulation of phosphatase activity		
		222.3–235.3	XM_037581542.1	Triticum dicoccoides xyloglucan endotransglucosylase/ hydrolase	Participates in cell wall construction of growing tissues. Involved in the accumulation of hemicelluloses, xyloglucan metabolic process, cell wall biogenesis, organization and macromolecule catabolic process		
Field tolerance to powdery mildew in two different locations during two years	6R	35.5–60.2				M	
		156.3–172.1					
		194.7–209.4					
**Drought tolerance in the adult stage (controlled conditions)**							
Cell wall-bound phenolics content in leaves under drought stress	4B	64.2–76.0					[34]
	3R	55.6–60.1	XM_044496729.1	General transcription factor IIE subunit	Transcription initiation from RNA polymerase II promoter		
	6R	331.2–345.9					
The maximum quantum efficiency of PSII (*F_v_/F_m_)* under drought stress	1R	147.5–150.9	U39321.1	Acetyl-CoA carboxylase	Multifunctional enzyme that catalyzes the carboxylation of acetyl-CoA, forming malonyl-CoA, which is used in the plastid for fatty acid synthesis and in the cytosol in various biosynthetic pathways including fatty acid elongation. Required for very long chain fatty acids elongation. Necessary for embryo and plant development. Plays a central function in embryo morphogenesis, especially in apical meristem development. Involved in cell proliferation and tissue patterning. May act as a repressor of cytokinin response.		
**Yielding capacity in the field**							
Straw height	5R	0.0–97.1					[38]
Spike length	6B	223.6–245.3	XM_044554178.1	Ethylene-responsive transcription factor ERN1	Transcription factor involved in the symbiotic nodule signaling pathway in response to rhizobial stimulation. Functions as a transcriptional regulator required for root infection by symbiotic rhizobia, infection thread (IT) formation, and nodule development. May coordinate these processes.		
Grains per spike	5A	223.1–232.6					
	2B	138.3–152.7	a. KR082547.1 b. XM_044538656.1	a. Cultivar Extra Early Blackhull FTSH protease 4-B1 b. Carotenoid 9,10(9',10')-cleavage dioxygenase-like	a. Acts as a processive, ATP-dependent zinc metallopeptidase for both cytoplasmic and membrane proteins. Plays a role in the quality control of integral membrane proteins. b. Involved in strigolactone biosynthesis, hormones that inhibit tillering and shoot branching, contribute to the regulation of shoot architectural response to phosphate-limiting conditions, and function as a rhizosphere signal that stimulates hyphal branching of arbuscular mycorrhizal fungi and trigger seed germination of root parasitic weeds.		
	5R	180.2–203.9					
Thousand kernel weight	5R	0.0–7.9					

* H—cv. ‘Hewo’; M–cv. ‘Magnat’.

**Table 2 plants-12-00619-t002:** Candidate genes in silico identified within 0–37 cM region of chromosome 5R in DH Hewo × Magnat winter triticale population.

No.	Gene Name	Predicted Function
1	Chloroplastic protein FAF-like *	Negative regulation of ABA-activated signaling pathway, negative regulation of phosphorylation, positive regulation of phosphatase activity
2	F-box/FBD/LRR-repeat protein *	Negative regulation of long-day photoperiodism and flowering; regulation of short-day photoperiodism and flowering
3	Myosin-10-like *	ATP binding myosin complex, cytoskeletal motor actin filament binding
4	Thylakoid membrane protein TERC *	Integral thylakoid membrane protein that plays a crucial role in thylakoid membrane biogenesis and thylakoid formation in early chloroplast development, essential for synthesis of photosystem II (PSII) core proteins and required for efficient insertion of thylakoid membrane proteins. May assist synthesis of thylakoid membrane proteins at the membrane insertion step
5	Xyloglucan endotransglucosylase/hydrolase *	Hydrolase activity, hydrolyzing O-glycosyl compounds; apoplast; cellular glucan metabolic process; xyloglucan: xyloglucosyl transferase activity
6	Nucleotide-gated ion channel *	Voltage-gated potassium channel activity, cyclic nucleotide gated ion channel, integral transmembrane helical component of membrane

*—Positive additive effect from cv. Hewo.

**Table 3 plants-12-00619-t003:** Other genes located in a section of chromosome 5R in rye database.

No.	Gene Name	Predicted Function [GO]
1	2-oxoglutarate (2OG) Fe(II)-dependent oxygenase (2x)	L-ascorbic acid binding, oxidoreductase activity EC:1.14.11.2, acting on paired donors, with incorporation or reduction of molecular oxygen.
2	2-oxoglutarate-dependent dioxygenase	Involved in the oxidation of jasmonate (JA), synthesized in response to attack by pathogens and herbivores, which triggers the activation of defense responses *via* the JA-mediated signaling pathway.
3	Basic helix-loop-helix (bHLH) DNA-binding superfamily protein	Negative regulation of innate immune response, regulation of pattern recognition receptor signaling pathway, regulation of transcription, DNA-templated, response to cytokinin, unidimensional cell growth.
4	Basic-leucine zipper (bZIP) transcription factor family protein	Cellular response to glucose stimulus.
5	Ankyrin repeat-containing protein	Involved in salt stress tolerance. May act through abscisic acid (ABA) signaling pathways and promote reactive oxygen species (ROS) production.
6	AP-2 complex subunit alpha-2	Clathrin-dependent endocytosis, intracellular protein transport, receptor-mediated endocytosis.
7	ATP-dependent RNA helicase DeaD	Involved in mRNA turnover, and more specifically in mRNA decapping. Involved in response to cold, salt stress and water deprivation.
8	Beta-amylase	Response to water deprivation and starch catabolic process.
9	BTB/POZ and MATH domain-containing protein (2x)	Protein ubiquitination and response to abiotic stimulus, a.o. salt stress, osmotic stress and water deprivation.
10	Carboxyl-terminal peptidase putative DUF239 (2x)	Neprosin, integral component of membrane.
11	Chaperone DnaK (2x)	ATP-dependent protein folding chaperone.
12	Cold regulated protein (COR) 27	Negative regulation of transcription, DNA-templated, regulation of circadian rhythm, regulation of photoperiodism, flowering, response to abscisic acid, response to absence of light, response to blue and red light, response to cold and virus, vegetative to reproductive phase transition of meristem.
13	Cyclin delta-3	G1/S transition of mitotic cell cycle, guard mother cell differentiation, regulation of cyclin-dependent protein serine/threonine kinase activity, response to cytokinin, response to sucrose, seed development.
14	Cysteine protease (4x)	Programmed cell death involved in cell development.
15	Cysteine proteinase inhibitor	Cellular response to heat, defense response, response to cold, water deprivation and oxidative stress, positive regulation of seed germination.
16	Cytochrome P450 (10x)	Leaf and root development, oxidoreductase activity, regulation of growth, response to insect, signalling.
17	Dihydrofolate reductase	Tetrahydrofolate biosynthetic process.
18	Dipeptidyl peptidase (2x)	Proteolysis.
19	Dirigent protein DNA (Cytosine-5-)-methyltransferase (3x)	Phenylpropanoid biosynthetic process.
20	Egg cell-secreted protein 1.1	Regulation of double fertilization forming a zygote and endosperm.
21	Embryogenesis transmembrane protein-like (2x)	Brassinosteroid mediated signaling pathway, cell death, cellular response to hypoxia, leaf senescence, negative regulation of cell death, protein phosphorylation, regulation of seedling development.
22	F-box family protein (5x)	Regulation of cell division, seed development, regulation of stomatal movement, response to abscisic acid, response to water deprivation.
23	Flavin-containing monooxygenase	Glucosinolate biosynthetic process from homomethionine, defense response to bacterium, defense response to fungus, L-pipecolic acid biosynthetic process, plant-type hypersensitive response, response to other organism, systemic acquired resistance.
24	Folylpolyglutamate synthase	Folic acid-containing compound biosynthetic process, photorespiration, seedling development.
25	Glycerol-3-phosphate acyltransferase 3	Fatty acid biosynthetic process, cutin and suberin biosynthetic process.
26	Glycolipid transfer protein domain-containing protein (3x)	Lipid transfer activity.
27	Glycosyltransferase (3x)	Regulation of plant-type cell wall cellulose biosynthetic process.
28	Heat shock transcription factor (3x)	Cellular heat acclimation.
29	Histone acetyltransferase of the CBP family 12 (7x)	Histone acetylation, positive regulation of transcription by RNA polymerase II.
30	Homeobox leucine-zipper protein	Auxin-activated signaling pathway, negative regulation of transcription DNA-templated, shade avoidance, unidimensional cell growth, reproduction, response to abscisic acid and cytokinin, osmotic stress, virus and water deprivation.
31	Kinase interacting (KIP1-like) family protein	Actin binding.
32	L-gulonolactone oxidase (2x)	L-gulonolactone oxidase activity, oxidoreductase activity, L-ascorbic acid biosynthetic process.
33	Lipid transfer protein (9x)	Defense response to fungus, induced systemic resistance, response to abscisic acid, response to cold, response to salt stress.
34	L-type lectin-domain containing receptor kinase VIII	Defense response to bacterium, defense response to oomycetes.
35	LURP-one-like protein	Might be related to the phospholipid scramblase and tubby-like superfamily of membrane tethered transcription factors.
36	Metacaspase-1	Defense response, positive regulation of programmed cell death, proteolysis.
37	Mucosal address in cell adhesion molecule 1	Positive regulation of cell population proliferation.
38	NAC domain-containing protein	Positive regulation of secondary cell wall biogenesis, regulation of transcription DNA-templated, defence response.
39	Nascent polypeptide-associated complex subunit beta	Polysomal ribosome, cold acclimation, response to salt.
40	Nicotianamine synthase (3x)	Synthesizes nicotianamine, a polyamine which serves as a sensor for the physiological iron status within the plant, and/or might be involved in the transport of iron.
41	O-methyltransferase (3x)	Methylation.
42	Organic cation transporter protein	Organic cation transport.
43	P1 (2x)	Aminopeptidase activity, manganese ion binding, N-1-naphthylphthalamic acid binding, zinc ion binding, auxin polar transport.
44	Peptidoglycan-binding LysM domain protein	Defence response, response to chitin.
45	Peroxidase (2x)	Hydrogen peroxide catabolic process, response to oxidative stress.
46	Photosystem I assembly protein Ycf3	Essential for the assembly of the photosystem I (PSI) complex. May act as a chaperone-like factor to guide the assembly of the PSI subunits.
47	Photosystem II CP43 reaction center protein	Photosynthetic electron transport in photosystem II.
48	Polyamine oxidase	Plays an important role in the regulation of polyamine intracellular concentration. Involved in abscisic acid-mediated developmental processes. May contribute to nitric oxide-mediated effects on root growth.
49	Polyphenol oxidase (3x)	Pigment biosynthetic process.
50	Polyubiquitin (2x)	Ubiquitin-dependent protein catabolic process, response to salicylic acid and UV.
51	Protease inhibitor/seed storage/lipid transfer protein	Lipid transfer activity.
52	Protein FAR1-RELATED SEQUENCE 5	Regulation of transcription, DNA-templated. Putative transcription activator involved in regulating light control of development.
53	Retrotransposon protein	Retrotransposon protein.
54	Ribosomal RNA small subunit methyltransferase J (2x)	Cell division and fate specification.
55	rRNA N-glycosidase (2x)	Defence response.
56	Senescence regulator (DUF584)	Senescence regulator (DUF584)
57	Small nuclear ribonucleoprotein	Post-transcriptional gene silencing by RNA, spliceosomal snRNP assembly.
58	Terpene synthase	Mono-/diterpenoid biosynthetic, response to herbivore, response to jasmonic acid, response to wounding.
59	THO complex subunit 1	mRNA export from nucleus and splicing, ta-siRNA processing, regulation of DNA-templated transcription, elongation.
60	Transcription elongation factor GreA	Regulation of DNA-templated transcription, elongation.
61	Transmembrane and coiled-coil domain-containing protein 4	Endoplasmic reticulum to Golgi vesicle-mediated transport.
62	Transmembrane protein (4x)	Transmembrane protein.
63	Transporter-related family protein	Transporter-related family protein.
64	Trihelix transcription factor GT-2	DNA-binding transcription factor activity.
65	Tryptophan decarboxylase	Serotonin biosynthetic process from tryptophan.
66	Tyrosine decarboxylase	Cellular amino acid metabolic process.
67	WD repeat-containing protein	Developmental vegetative growth.
68	Xyloglucan 6-xylosyltransferase (2x)	Xyloglucan biosynthetic process, cell wall biogenesis/degradation.

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
