# Peer review of "The Genome Regions Associated with Abiotic and Biotic Stress Tolerance, as Well as Other Important Breeding Traits in Triticale"

_plants, 2023, doi:10.3390/plants12030619_

Round 1

Reviewer 1 Report

The authors describe well review paper regarding genetic linkage map, quantitative trait loci and candidate genes in Triticale.  Genome information on both low temperature and snow mold tolerances which are important traits would be useful information in breeding program of Triticale.

 However, overall, paper feels like a mishmash of data, so it needs to be more summarized.  Note the consistent use of abbreviations in a text.

L39  Genome constitution  AABBRR

L69  Markers should be spelled out.

 Section 9 is missing.  Any other trait?  Or A simple mistake in numbers.

 In Section 4 and 5, authors mentioned the characteristics of both low temperature and snow mold tolerances.  I didn't think it was necessary to describe them in such detail for the purpose of this paper.

L 159   M. nivale   spell out

Author Response

Dear Reviewers, we are very grateful for your valuable comments. We are sending back the manuscript in which we have taken into account all your suggestions.

General amendments made:

  1. Each figure and table were put after the first mention in the revised manuscript.
  2. All references were check in the relevanceto the contents of the manuscript.
  3. All revisions to the manuscript text were marked up using the “Track Changes” function in MS Word.
  4. A cover letter is provided below to explainthe details of the revisions to the manuscript and responses to your
  5. English language and style were checked.

Reviewer 1 Comments and Suggestions for Authors:

The authors describe well review paper regarding genetic linkage map, quantitative trait loci and candidate genes in Triticale. Genome information on both low temperature and snow mold tolerances which are important traits would be useful information in breeding program of Triticale.

However, overall, paper feels like a mishmash of data, so it needs to be more summarized. Note the consistent use of abbreviations in a text.

The work has been improved in terms of the arrangement of subsections, as well as in terms of linguistic clarity. Figures and tables have been added where appropriate.

L39  Genome constitution  AABBRR

Line 39 describes a first triticale hybrid that was created, which originally was an octoploid with AABBDDRR genome. Currently cultivated triticale has no DD genome (genome D is eliminated during a breeding process) and a proper construction of genome (AABBRR) was added to the line ‘…hexaploid triticale contains genomic constitution of 2n = 6x = 42 with AABBRR genome’.

L69  Markers should be spelled out.

Markers were spelled out: ‘Most of those maps were developed using DArT (Diversity Arrays Technology), DArT-seq (Diversity Arrays Technology Sequencing) and SNP (Single-nucleotide Polymorphism) marker systems which are widely used in genetic map construction for multiple crop species’.

Section 9 is missing. Any other trait? Or A simple mistake in numbers.

It was mistake in chapter’s numbering. It was corrected.

In Section 4 and 5, authors mentioned the characteristics of both low temperature and snow mold tolerances. I didn't think it was necessary to describe them in such detail for the purpose of this paper.

For the purposes of the review article and the requirements regarding the minimum number of characters, we have decided to leave a separate subchapter on tolerance to M. nivale infection, because it seems to us important due to the current threats to triticale crops as well as due to the number of tests carried out on the described main population of the DH ‘Hewo’ x ‘Magnat’ lines.

L 159 M. nivale spell out

That was corrected.

Yours sincerely,

Gabriela Golebiowska-Paluch

Reviewer 2 Report

1. When talking about the triticale genome the terminology "subgenomes A, B, R, D" is usually used, but not "part of the triticale genome" and not "genome A", "Genome B", etc. or "wheat A genome".

2. The key for comparison is the population of DH Evo x Magnat to which there are comments - it is quite limited in number - 89 lines only and markers are associated in 20 linkage groups instead of 21. How correct is it to use this population as a basis?

3. The text of the article does not focus on the very different results of two similar experiments on seedlings tolerance to M. nivale including the chromosomes of the  R subgenome.

4. Table 1 (rows 298-306) presented without a name to be insufficiently elaborated. There is a lot of information directly having a dubious connection with the described signs, or is indicated without connection with anything. It is necessary to think about a more convenient perception of the presented information.

Author Response

Dear Reviewers, we are very grateful for your valuable comments. We are sending back the manuscript in which we have taken into account all your suggestions.

General amendments made:

  1. Each figure and table were put after the first mention in the revised manuscript.
  2. All references were check in the relevanceto the contents of the manuscript.
  3. All revisions to the manuscript text were marked up using the “Track Changes” function in MS Word.
  4. A cover letter is provided below to explainthe details of the revisions to the manuscript and responses to your
  5. English language and style were checked.

Reviewer 2 Comments and Suggestions for Authors:

  1. When talking about the triticale genome the terminology "subgenomes A, B, R, D" is usually used, but not "part of the triticale genome" and not "genome A", "Genome B", etc. or "wheat A genome".

It was corrected in our revised manuscript.

  1. The key for comparison is the population of DH Evo x Magnat to which there are comments - it is quite limited in number - 89 lines only and markers are associated in 20 linkage groups instead of 21. How correct is it to use this population as a basis?

Many different experiments on the DH ‘Hewo’ x ‘Magnat’ lines population were performed. But publishing the genetic map for this population which was in 2015 allow us to introduce a new set of analysis. We are aware that 89 lines of this population can be considered as poor number of lines but we have managed to identify a QTL and candidate analysis associated with multiple traits that have been already published over last few years (GoÅ‚Ä™biowska et al. 2021, WÄ…sek et al. 2022, Dyda et al. 2022).

Regarding 20 linkage groups of this mapping population, chromosome 7R was eliminated from this genetic map as markers segregation of this chromosome was inconsistent and was associated with the introduction of high distance (over 30 cM) between markers on this chromosome.

  1. The text of the article does not focus on the very different results of two similar experiments on seedlings tolerance to nivaleincluding the chromosomes of the  R subgenome.

In the improved Table 1 above information were clarified. The main discussion of these results was made in the original experimental paper (Gołębiowska et al. 2021). Its most important elements have been introduced to the summary of the revised review article in the subchapter Conclusions.

  1. Table 1 (rows 298-306) presented without a name to be insufficiently elaborated. There is a lot of information directly having a dubious connection with the described signs, or is indicated without connection with anything. It is necessary to think about a more convenient perception of the presented information.

The improved Table 1 has been inserted at the appropriate place in the text. The header and descriptions of rows and columns are visible. Unnecessary abbreviations and numbers have been replaced with the full name of the feature.

Yours sincerely,

Gabriela Golebiowska-Paluch

Round 2

Reviewer 1 Report

non